# Optimization of REBCO Tapes through Division and Striation for Use in Superconducting Cables with Low AC Losses

**DOI:** 10.3390/ma16237333

**Published:** 2023-11-25

**Authors:** Marcela Pekarčíková, Lubomír Frolek, Martin Necpal, Eva Cuninková, Michal Skarba, Simona Hulačová, Filip Ferenčík, Barbora Bočáková

**Affiliations:** 1Faculty of Materials Science and Technology in Trnava, Slovak University of Technology in Bratislava, Jána Bottu 2781/25, 917 24 Trnava, Slovakia; martin.necpal@stuba.sk (M.N.); eva.cuninkova@stuba.sk (E.C.); michal.skarba@stuba.sk (M.S.); simona.hulacova@stuba.sk (S.H.); filip.ferencik@stuba.sk (F.F.); barbora.bocakova@stuba.sk (B.B.); 2Institute of Electrical Engineering, Slovak Academy of Sciences, Dúbravská Cesta 9, 841 04 Bratislava, Slovakia; elekfro@savba.sk

**Keywords:** high-temperature superconducting tapes, AC loss, striation, laser ablation, cutting, superconducting cable

## Abstract

This study aimed to enhance the performance of Ag-stabilized high-temperature superconducting (HTS) tapes with a focus on reducing magnetization losses. Two approaches were employed: dividing the tapes into narrower widths and introducing striation at the level of the superconducting layer. The process of laser ablation proved to be an effective method for implementing these modifications. The quality of the cut edges and grooves was assessed using scanning electron microscopy. To evaluate the electrical properties, measurements were conducted on the critical current and magnetization loss in samples at different stages: in their initial state, after cutting, and after the striation process. Of the two modifications, the striation process more effectively reduced the AC losses in the HTS tapes, approximately by one order of magnitude. The retention of critical current remained high after cutting, but varied with the number of created filaments after the striation process. Subsequently, a short cable was wound from the cut and striated HTS tape. This cable demonstrated a remarkable sixfold reduction in AC losses compared to the initial HTS tape.

## 1. Introduction

REBCO (RE—rare earth, B—barium, C—copper, O—oxygen)-based high-temperature superconductors (HTS) have great potential in various applications, such as transmission cables, generators, motors, transformers, fault current limiters, and magnets [1,2]. They offer increased efficiency and reduced operating expenses. Currently, these superconductors are manufactured in the form of approximately 0.1 mm thin coated conductor tape with different widths (up to 12 mm) and lengths exceeding hundreds of meters, which is able to transport electrical currents in the 100–1000 A range when cooled to 77 K using liquid nitrogen (LN_2_) [3,4,5,6,7,8,9]. In HTS tape, the superconductor layer deposited on the Hastelloy substrate is very thin, typically about 1–3 µm. This results in an enormous thickness-to-width aspect ratio ~1:10,000, which makes the REBCO superconductor susceptible to high-power dissipation whenever there is a change in the magnetic field, leading to significant alternating current (AC) magnetization losses.

The pulse magnets used in fusion reactors and particle accelerators are a potential application of REBCO superconductors [10,11]. However, in fusion or accelerator magnets, the current needs to be ramped from zero to the final value, exposing the HTS tape to varying magnetic fields and resulting in the aforementioned power dissipation, which hinders its practical application. Paradoxically, it is the pinning of magnetic flux, the mechanism that secures the transportation of electrical current without resistance in type II superconductors, that causes magnetic hysteresis, resulting in the dissipation of energy [12]. According to Brandt and Zeldov’s theories [12,13], the value of magnetization loss is approximately proportional to the width of the superconducting layer. Therefore, an obvious way to reduce the magnetization loss is to produce narrower HTS tapes or segment the wide superconducting layer into filaments.

The operating currents required in pulse magnet cables cannot be reached using a single HTS tape. This problem necessitates assembling several tapes into wires, which are then bundled to form a cable capable of carrying electrical currents exceeding several tens of kiloamperes. Currently, 4 mm, 3 mm, and 2 mm tape widths are used for assembling superconducting cables by winding the flat tape around a round core in several layers with a spiral-shaped arrangement [14,15]. Although HTS tapes with such widths have a lower critical current, *I_c_*, using sufficiently thin substrates (30 µm) allows the production of very flexible wires known as CORC^®^ (Conductor on Round Core) wires [16]. Narrower tape widths induce smaller stresses and allow easier sliding during cable winding. The irreversible strain limit of REBCO tape under axial compression is about twice as high as under axial tension [17]; therefore, the HTS layer should be positioned on the inner side of the helix, where more advantageous compressive stresses act [18]. Another important factor that influences the performance of the tape in CORC^®^ cables is their winding angle, which should be approximately 45° due to the anisotropic in-plane reversible strain effect of the REBCO tapes [19].

From an economical point of view, it is not a common practice to deposit the buffer and REBCO layers on very narrow substrates. Generally, a 12 mm wide substrate tape is used, which is then slit into narrower tapes after coating it with the Ag layer. However, mechanical dividing using a shearing mechanism, previously used routinely in production, is not recommended for achieving narrow tapes, as it can cause cracking of the ceramic superconductor within several tens of micrometers from the edge of the tape [18]. With a tape width narrower than 2 mm, this can lead to a reduction in the superconducting area in its cross-section by up to 10–20%, and the existing cracks may initiate undesirable delamination of the REBCO layer. Therefore, there is a need to approach new HTS-tape-cutting possibilities. Laser cutting is one option for continuous division without blunting the cutting tool. Though it carries the risk of thermally affecting the cut edges, more and more manufacturers are switching to this cutting method. Surprisingly, only a few peer-reviewed publications are devoted to this topic in the literature. Gu et al. [20] used an ultraviolet laser device with a relatively low power level of 3 W to cut 150 µm thick HTS tapes. The group introduced a reel-to-reel system for processing long-length HTS tapes. Heilmann et al. [21] also employed the reel-to-reel cutting process, but they used a water-jet-guided laser instead of a scribing laser ablation technique to reduce the heat-affected zone and to efficiently remove the molten material using a water jet. They set the laser power to 17 W for cutting the HTS tapes with a thickness of 55 µm. Thin REBCO single crystals (15–30 µm) were cut without damaging the superconductor in another work [22] using a diode-pumped Nd:YLF laser with a low mean power of 135 mW. The published results were achieved using parameters that are not sufficiently described and difficult to follow.

Producing tapes narrower than 1 mm would probably be inefficient. However, sufficiently narrow tapes can be further segmented at the superconducting layer level using the so-called striation process [23]. The striated structure should have slots between the adjacent filaments that reach the metallic substrate, preserving the high mechanical strength of the HTS tape. Various methods have been tested by different groups to create such a structure, including the dry etching process [24], the laser scribing method [25,26,27], or the laser scribing method combined with wet chemical etching [28,29]. These methods have been proven to reduce AC losses, and in general, the *I_c_* retention was higher than the theoretical value calculated based on the removed superconducting material, particularly when the filaments became thinner than 300 µm.

Filaments reduce the hysteretic magnetization AC losses, but at the same time, another kind of dissipation appears in HTS tape: a coupling loss caused by the currents induced in the loops that connect the filaments covered by Cu stabilization. To suppress the coupling losses in striated and Cu-stabilized HTS tapes exposed to AC fields, the conductor needs to be transposed along its length, either by twisting the tape or winding it helicoidally [30,31,32]. On the other hand, a certain extent of inter-filament coupling is desirable in a technical conductor for good current sharing and stabilization.

In this paper, we applied both approaches to reduce the AC loss in HTS tapes: dividing and striating Ag-stabilized HTS tapes. Saw blades designed to cut the Ni superalloys and laser ablation were used for the dividing process. Laser technology was also employed to create striations in the HTS tapes. Subsequently, the cut and striated HTS tape was wound into a short one-layer cable model. At each preparation step, we conducted electrical characterization, measuring the *I_c_* and AC loss for the samples.

## 2. Materials and Methods

Two commercially available HTS tapes were used for the experiments, purchased from the S-Innovations (Moscow, Russia) and SuperPower (Furukawa Electric Group, Glenville, NY, USA) companies (hereinafter referred to as SI and SP, respectively). Both tapes were 12 mm wide and had only Ag stabilization, but differed in their substrate thickness and the thicknesses of particular deposited layers, as shown in Figure 1. The roughness of the Ag layer in the SP tape was much higher than that in the SI tape.

At first, the as-purchased samples were characterized in their initial state using *I_c_* and AC loss measurements. In order to ensure optimal electrical contact during the *I_c_* measurements, a layer of electroplated copper was added to both ends of the 10 cm long samples, each covering a 2.5 cm length. The self-field *I_c_* was measured in a LN_2_ bath using the four-probe method, adhering to the internationally agreed electric field criterion of 1 μV/cm. For the AC measurements, samples with a length of 5 cm were used, and copper contact areas were not produced, as they could influence the measurements. Consequently, if the sample had ends with copper terminals, they were cut off. The magnetization losses were measured at 77 K under the effect of an applied external sinusoidal magnetic field. The specific energy, *Q*, dissipated in each cycle of applied magnetic field and per unit length of the HTS tape samples was used to quantify the magnetization losses. The applied magnetic field, *B*, varied from 1 to 100 mT, with discrete frequencies, *f*, of 36, 72, and 144 Hz. We utilized a specific so-called calibration-free method for these measurements. This method is based on the measurement of the part of the power which is supplied by the AC source to the AC magnet generating the magnetic field, in which the sample is located. It uses a coil wound in parallel with the AC field magnet as the measurement coil. To ensure the required sensitivity, we employed two identical systems, each comprising an AC magnet and a measurement coil. One of these setups contained the sample, while the other remained empty. This method eliminates the need for any calibration constant measurements and is extensively detailed in [33].

The Hastelloy substrate, used for the measurement of AC loss, was obtained from the original S-Innovations tape by dissolving the Ag and REBCO/buffer layers in KI_3_ and HNO_3_ etching agents, respectively. The electrical measurements were repeated after modifying the samples using the cutting or striating processes.

The 12 mm wide HTS tapes were divided into three pieces along the tape length using diamond and cubic boron nitride (CBN) cutting saw blades (Buehler, Lake Bluff, IL, USA) or using laser ablation. The mechanical cutting was carried out using a precision saw ISOMET 5000 (Buehler, Lake Bluff, IL, USA). The tapes were placed between two flat epoxy resin boards to prevent slipping as well as bending of the tape during cutting. A water-based coolant (Cool 3) was applied to cool the samples and prevent overheating. The thickness of the cutting blade was 0.6 mm. The cutting parameters varied in the feed rate, ranging from 1.2 to 10 mm/min, and in the number of rotations per minute (RPM), ranging from 3000 to 5000.

A LASERTEC 80 Shape fiber laser machine (DMG Mori, Stollberg, Germany) was used to cut the HTS tape using laser ablation. This machine was equipped with a fiber ytterbium laser source with a wavelength of 1068 nm. It operates in a 120 ns long pulse regime. The frequency of pulses varied between 80 and 100 kHz. The scanning speed of the laser beam was adjusted in the range of 1000 to 2000 mm/s. The power of the laser generator was selected between 48 and 100 W. During the cutting, various liquids and gases such as water, oil, argon, and compressed air were tested for better dissipation of the generated heat. The cooling gas stream was directed to the cut location at a flow rate of 20–30 L/min. In case, where a liquid cooling medium was used, the sample was immersed in a liquid bath, ensuring that the liquid exceeded the sample’s surface by approximately 1.5 mm. The samples were cut from both sides of the HTS tape, either from the top (REBCO) side or the bottom (Hastelloy) side. The specific cutting parameters used for particular samples are listed in Table 1. Similar to the mechanically cut samples, these samples were also divided into three pieces with approximately the same width of 4 mm.

The same laser machine was used to create filaments in the HTS tapes. For this process, significantly smaller values in the process parameters were employed compared to cutting. The power, frequency, laser beam speed, and number of repetitions varied within the ranges 0.6–10 W, 10–20 kHz, 1000–500 mm/s, and 1–2 cycles, respectively. Compressed air was used as the coolant. The specific process parameters used for creating the filaments in the samples are summarized in Table 2. The striation pattern consisted of 8, 15, or 30 filaments, and it was created across an area of 50 mm × 12 mm in the middle of the 100 mm long sample.

The edge quality after the cutting or striating process was checked using a scanning electron microscope (SEM) JEOL JSM-7600F (JEOL, Tokyo, Japan) equipped with secondary electron (SE) and back-scattered electron (BSE) detectors and X-ray energy-dispersive spectrometry (EDX). The examination was carried out after etching the Ag layer using an aqueous solution of KI_3_.

As a result of the cutting and striating processes, a certain amount of HTS material was removed. This loss was considered when determining the *I_c_* retention. The calculation of material loss relied on SEM measurements of the groove and cut widths. The grooves were directly measured on the samples under assessment. To determine the widths of the cuts, we employed shorter samples that were only partially divided (cut halfway). This approach ensured that the cut sections remained in place without shifting, enabling precise measurement of the cut width. The cutting process parameters applied to these shorter samples were identical to the samples being evaluated.

In the final step, a single-layer cable model was created using a SI tape that had undergone both cutting and striating processes. This modified tape was prepared using laser ablation, employing the process parameters designed for the samples SI-CA2 (cutting) and SI-S2 (striating). It had a width of 2 mm and featured three filaments, each measuring approximately 670 µm in width. To safeguard the exposed superconductor within the formed grooves, a layered protective system 30 nm Ti/1 µm AlN/30 nm Ti/150 nm Ti-Cu/500 nm Cu was applied to the top of the sample using magnetron sputtering. The 10 cm long HTS tape was then helically wound in two complete turns around an electrically insulating former with a 7 mm diameter, which was crafted from polyethylene terephthalate glycol-modified material with carbon fibers (PETG CF). This composite former was designed in the form of a flat spring, providing sufficient flexibility. It was created using 3D-printing technology. The choice of the insulating former aimed to minimize any additional magnetization losses, and further details about its development can be found in [34,35]. Throughout the winding process, the tape was maintained at a 45° winding angle, and a tensioning force of 2.5 N was applied. The REBCO layer was positioned on the inside of the helical arrangement. Following the steps of cutting, striating, and applying the protective layer, the HTS tape was subjected to *I_c_* measurements. Subsequently, once it was wound into the cable configuration, the *I_c_* measurements were repeated, along with the AC loss measurements.

## 3. Results

### 3.1. Characterization of HTS Tapes in the Initial State

Measuring the critical current values of the HTS tapes in their pristine state is of the utmost importance. Comparing these initial values with those obtained after the cutting and striating processes allows us to assess any potential degradation that may have occurred. Anticipated is a reduction in *I_c_* due to inevitable losses in the superconducting material during the modification procedure. However, the damage inflicted on the cut edges during the cutting process should be minimized to the greatest extent possible. Figure 2 displays the typical DC voltage-current (*V*-*I*) characteristics of the investigated HTS tapes in their initial state. This measurement was performed in a self-field at 77 K. For the SI tape, the average *I_c_* value was measured at 471 ± 9 A, with an accompanying *n*-value indicating the steepness of the transition into the superconducting state, which was 38 ± 2. The SP tape exhibited an *I_c_* of 367 ± 14 A and an *n*-value of 32 ± 2. These differences can be attributed to the SP tape’s thinner REBCO layer.

Figure 3a presents an illustrative example of the results for the AC losses at a frequency of 36 Hz measured in the initial state of the HTS tapes. The trend of the *Q*(*B*_rms_) dependence, where the AC losses increase with the third power, typical for superconductors, remained highly consistent across all the measured samples, particularly when the magnetic fields exceeded 1 mT. In Figure 3b, the measurements of the same samples at three different frequencies exhibited a similar pattern. Additionally, this graph includes measurements of the AC losses for the pure Hastelloy substrate (referred to as Hast) that rise with the square power, as is characteristic for conductive materials. It is evident that the noise recorded at the beginning of the curve for the HTS tapes aligns with the measurement for the pure Hastelloy substrate (prepared from the SI tape). This observation underscores the overall uniformity of the electrical properties of both HTS tapes, making them well-suited materials for modification and subsequently cable production with low AC losses.

### 3.2. Modification of HTS Tapes by Narrowing the Superconductor Width

#### 3.2.1. Cutting of HTS Tapes

In the case of mechanical cutting, it became evident that the quality of the cut edge is notably influenced by the type of HTS tape used. While the SuperPower tape can be cut without any visible damage, the Ag stabilization layer of the S-Innovations tape delaminates considerably during cutting, as shown in Figure 4a. This disparity could be attributed to the fact that the S-Innovations tape has a thinner substrate, making it less robust. Furthermore, the delamination of only the Ag layer also indicates a potentially weaker adhesion to the superconductor surface compared to the SuperPower tapes. Consequently, our decision was to apply mechanical cutting exclusively to the SuperPower tapes.

The choice of cutting blade type significantly impacts the quality of the cut edges. Employment of a diamond blade produces voluminous metal turnings at the cut edges, as visible in Figure 4b. According to the EDX analysis, they consist of ductile silver and a Hastelloy alloy. When the HTS tapes are wound helically around a circular former, the tape edges are raised as a consequence of the Poisson effect. This causes indentations in the tapes in the subsequently wound layer of the cable [36]. Tape edges with additional turnings may introduce undesirable damage into the superconducting layer when the tapes are coiled. We encountered similar challenges in our previous work, where relatively small protrusions at the edges of the tape led to the cracking of the superconducting layer of the tape wound in the second layer of the cable.

When utilizing the CBN blade for cutting, the cut edge of the SuperPower tape was noticeably smoother, and there was no formation of metal turnings, as illustrated in Figure 4c. The CBN cutting blade used in this study is specifically recommended by its producer for use with Fe, Co, and Ni alloys, and superalloys. Given that the Ni superalloy is essentially the carrier component of the superconducting tape, we attribute the improved result to this compatibility. Furthermore, we examined the influence of cutting parameters such as RPM and the sample speed (feed rate) during cutting. However, no fundamental differences were observed in the settings of these various parameters concerning the quality of the cut edges. From a practical perspective, faster feed rates are preferred. All the samples shown in Figure 4 were cut at 3000 RPM and with a high feed rate of 10 mm/min.

Next, we examined the cut edges at the level of the brittle superconducting layer to assess the presence of cracks. For this purpose, the silver layer was removed using etching, as illustrated in Figure 4d. SEM observation revealed that no cracks were observed in the superconducting layer. However, at a higher magnification, it became apparent that the cut edge was not perfectly smooth, and there were remnants of deformed material from the ductile substrate. To achieve a smoother edge, the cut edge can be further polished using very fine grinding paper.

For dividing HTS tapes using laser ablation, we used a laser machine with a fixed pulse duration of 120 ns. Such extended pulses, when combined with maximal power, can locally heat the HTS tape to elevated temperatures. In this scenario, it is possible that the treated area does not have sufficient time to cool down between pulses, potentially having a destructive effect on the superconductor over a broader region [37]. We believe that a device with a shorter pulse duration, in the order of femtoseconds, would be more suitable for this purpose. Extremely short pulse durations are less likely to cause serious damage to the superconductor, even when employing high power, thereby allowing the tape to be divided in a single cutting cycle. Regrettably, due to the fixed long pulse duration of our current laser machine, we were compelled to offset this limitation by employing lower power settings and conducting the laser ablation process multiple times to cut through the entire thickness of the HTS tape. During this process, we applied various cooling media (water, oil, argon, or compressed air) to mitigate the heating effects. Examples of the cuts made under these conditions are shown in Figure 5.

Water cooling was found to be ineffective. Even with the highest laser power or 100 repetition cycles, it was not possible to divide the HTS tape. The visible expansion in width of the trace after laser ablation (as seen in Figure 5a), without a corresponding increase in depth, led us to conclude that this expansion was likely due to the scattering of the laser beam in the water. Better results were achieved with oil cooling (Figure 5b), but it still required a large number of ablation cycles to cut through the entire thickness of the sample. Moreover, the oil cooling medium seems to be inappropriate for cutting long tapes, as the ablated material accumulates in the oil, rapidly changing its optical properties and causing the laser beam to scatter.

Cooling of the cut area using a gas stream offers the advantage of a simpler implementation. Figure 5c,d provide examples of laser ablation cuts using argon and a compressed air flow, respectively. In both cases, lower power settings and fewer repetitions are sufficient for cutting compared to with liquid cooling media. However, one drawback is that the material removed from the cut is deposited on the surface of the cut edges, as shown in Figure 6a. EDX analysis performed on the cut side reveals elements from the Ag layer and the substrate (Figure 6c), and indicates also the formation of oxides due to the presence of a high content of oxygen. When no cooling is applied, more significant redeposition occurs. As a result of repeating the process, the cuts have a trapezoidal cross-section. Therefore, performing laser ablation on the HTS tape from the side of the substrate is more advantageous, as it removes a smaller amount of material from the superconducting layer. On the reverse side, there is also a minimal redeposition and oxidation, as can be seen in Figure 6b. The images were captured using the BSE detector to enhance the material contrast.

#### 3.2.2. Electrical Properties of Cut HTS Tapes

Based on our microscopic observations, we opted to assess the electrical properties of the samples prepared using mechanical cutting with the CBN blade and laser ablation without the use of any coolant and with cooling using a compressed air flow. The latter method appears to be the most practical for a reel-to-reel cutting process with integrated cooling. The results are presented in Figure 7.

When using the CBN blade for mechanical cutting at 5000 RPM and a feed rate of 10 mm/min, a cutting edge of visual high quality was achieved, devoid of any indications of superconductor cracking. However, electrical measurements unveiled a significant deterioration in *I_c_*, exceeding 26%. This degradation is likely attributed to the direct contact of the superconducting material with the water-based coolant. Additionally, the mechanical cutting method results in notable wastage of HTS material, as the cutting blades are relatively thick, leading to an approximately 8% loss in our scenario. Taking into account the removed superconducting material, the overall results remained dissatisfactory, with less than 82% *I_c_* retention with regard to the lost HTS material.

Cutting using laser ablation without any cooling led to an approximately 10% increase in *I_c_* retention compared to mechanical cutting. Introducing cooling using compressed air further improved the *I_c_* retention by about 5%. The distinction in *I_c_* retention between the cut tapes from the REBCO or substrate side did not show a clear trend, suggesting it has an insignificant role in *I_c_* retention. Of greater significance is the thickness of the HTS tape, which directly correlates with the number of cutting cycles required. For instance, the SP tapes were cut using 20 repeated cycles, while the thinner SI tapes needed only 13 cycles. This difference in cutting cycles led to an approximately 2% variance in the final *I_c_* retention. It appears that multiple instances of laser exposure in the same location affect a wider superconducting area, influencing the overall performance.

Considering the notably improved *I_c_* retention using laser cutting, we conducted measurement of the AC losses on two such samples, labeled SP-N2 and SI-CA2 in Figure 7. Both samples were cut from the Hastelloy side. The energy release profiles in response to magnetic field variations at three distinct frequencies for these divided tapes are presented in Figure 8. The AC losses of the same tapes in their original state are also included in the plots. The difference between the original and reduced-width tapes is approximately one order of magnitude. However, this comparison overlooks the varying *I_c_* values of the cut tapes, which were approximately 3.15 times lower than the *I_c_* of the original 12 mm wide HTS tape due to division into three pieces. Consequently, the measured AC losses in the narrowed tapes are not actually 10 times lower, as indicated in Figure 8. This discrepancy arises because the drop in *I_c_* in these tapes needs to be taken into account. In the final section, we will revisit these measurements and incorporate them in a comparable plot with normalized values. For magnetic fields where *B* > 3 mT, the measured data correlate well with the theoretical curves calculated based on the Brandt–Indenbom model, as utilized in the work by Shigemasa et al. [38]. Both graphs representing samples with an 8% difference in *I_c_* retention exhibit striking similarities, indicating that low *I_c_* retention has a minimal influence on AC losses, or it may not by detectable using our measuring system. Nevertheless, our aim remains to obtain HTS tapes with low AC losses while maintaining the highest attainable critical current.

#### 3.2.3. Striation of HTS Tapes

Creating grooves that defined the size of the filaments did not demand high power in the laser device; however, lower powers were significantly impacted by the roughness of the Ag cover layer. While power up to 6 W proved to be effective in producing grooves on the rougher SuperPower tapes, the laser beam appeared to act as if it was reflected by the smooth surface of the S-Innovations tapes, resulting in no interaction with the surface. Consequently, the power level was increased to 10 W. Examples of the grooves created in the HTS tapes of different roughness are presented in Figure 9a,b.

The grooves must have a sufficient depth that reaches at least the buffer layer. This necessitates a minimum of two cycle repetitions when using 6 W of power. The SEM images of the grooves in Figure 9 are supplemented with EDX maps representing selected elements from the substrate (Ni), buffer (O), superconducting (Ba, O), and stabilization (Ag) layers. While barium is absent in the groove created by two cycles, it remains present in the groove created by one cycle (Figure 9c). It is conceivable that the rest of the superconductor may be damaged and the groove may act as non-conductive. However, due to the challenge of achieving perfect tape flatness during the striation process, there is a risk that the groove could become shallower, resulting in an uninterrupted superconductor between the filaments. This indeed occurred in one of our samples striated with grooves shown in Figure 9c. Consequently, two repetition cycles were deemed suitable for the striation of HTS tapes intended for electrical measurements.

The width of the grooves depends on both the number of repetition cycles as well as on the power used. At lower powers, the groove width increases as the power is raised. Subsequently, it stabilizes within a range of 50–60 µm, as depicted in the graph in Figure 10. As previously mentioned, excessively low power settings fail to provide sufficient interruption of the superconducting layer along the tape length. It is worth noting that there was also some enlargement of the groove width after the second repetition due to a 10 µm displacement of the laser path from the trajectory of the initial cycle.

#### 3.2.4. Electrical Properties of Striated HTS Tapes

Figure 11a illustrates the results of the *I_c_* retention measurements for two different HTS tapes, both subjected to laser ablation striation. In these samples, eight filaments were created, separated by seven grooves. The removal of superconducting material from 60 µm and 55 µm wide grooves for the SI and SP tapes was calculated to be 3.5% and 3.2%, respectively. Remarkably, the *I_c_* retention, with consideration of the lost HTS material, is nearly 100% for the SuperPower tape. Similarly, the application of a 4 W higher power for striating the S-Innovations tape did not significantly impact its *I_c_*. The disparity in *I_c_* retention between them is just 1.8%.

Subsequently, SuperPower tapes with an increased number of filaments (15 and 30) were prepared and measured. However, the *I_c_* retention of the HTS tapes exhibited a linear decline with an increasing number of filaments, as depicted in Figure 11b. This decrease persists even when accounting for the lost superconducting material. The phenomenon of linear *I_c_* degradation after laser striation with an increasing number of filaments was also observed by Vojenčiak et al. [31]. Their study used stabilized HTS tapes with a 20 µm thick Cu layer applying a slightly higher laser power (12.5 W) along with significantly more repetition cycles per groove (50) compared to our striation process parameters. For their samples, the *I_c_* retention was 92% for the sample with 3 filaments, 65% for the sample with 7 filaments, and only 40% for the sample with 10 filaments. Another work by Demenčík et al. [39] presents results from similar HTS tapes, albeit stabilized using a 5 µm Cu layer. They created three, five, and seven filaments using about half the power and repetition cycles as those in [31]. The *I_c_* retention improved to approximately 87% for the sample with seven filaments.

Considering the high *I_c_* retention rates, it is thus more favorable to striate HTS tapes with the thinnest possible stabilization layers. This approach allows the use of lower power and in particular a maximum of two repetition cycles. These parameters can be linked to the amount of generated heat, which can potentially harm the superconductor. Additionally, further *I_c_* degradation occurs in striated HTS tapes as the filaments become thinner. In cases where structural inhomogeneities exist within the tape, causing a localized reduction in *I_c_*, it becomes more likely that one or more filaments will experience significant drops in their transmission capacity. In our samples, the filament widths were approximately 1600 µm, 750 µm, and 320 µm for 8, 15, and 30 filaments, respectively. A width of 320 µm could be critical for localized regions with a reduced *I_c_*.

All the striated samples depicted in Figure 11 were subjected to AC loss measurements at three district frequencies, and the results are presented in Figure 12. For comparative purposes, the AC losses of the SuperPower tape in its original state are also incorporated into the plots. Additionally, the theoretical curves, calculated based on the Brandt–Indenbom model, are included in the graph. The difference between the original (non-striated) tape and the tapes containing eight filaments is approximately one order of magnitude, as clearly depicted in Figure 12a. Furthermore, Figure 12b demonstrates that as the number of filaments increases, there is a further decline in the AC losses.

In regions characterized by a low magnetic field, particularly when *B* < 0.5 mT, the magnetization losses exceed the theoretical expectations. This phenomenon primarily stems from the coupling losses that occur between the filaments. The observed increases in magnetization losses, especially within the low-field region, can also be attributed to the possible degradation of the critical current near the edges of the HTS tapes, as reported in [40]. In our striated samples, several tens of edges were created at the level of the superconductor layer, particularly when 30 filaments were produced. On these edges, the superconductor can be significantly affected by heat generation during laser cutting or scribing. A larger deviation from the theoretical values at lower magnetic fields, visible in Figure 12b, may also be associated with this phenomenon.

#### 3.2.5. Cut and Striated HTS Tape Wound on Cable Former

The electrical characteristics of the fabricated short single-layer cable were examined. The HTS tape used for this cable was modified using laser ablation, resulting in 2 mm wide tape with three filaments. The cutting process parameters were identical to those applied to sample SI-CA2, and the striating parameters matched those of sample SI-S2. As a result of this procedure, the tape exhibited an *I_c_* retention of 76.5 A, which is quite satisfactory, as it closely approaches the anticipated theoretical value of 78.5 A, calculated by dividing the *I_c_* value of the original 12 mm wide tape by six.

Upon winding this tape onto a former with a 7 mm diameter, both the *I_c_* and *n*-value decreased by about 12% and 16%, respectively. This reduction is likely attributed to the introduction of strain into the tape resulting from the winding process. Another potential factor contributing to this degradation, which we cannot yet rule out, is the possibility that the protective layer system deposited on top of the striated HTS tape did not adequately shield the superconductor within the grooves. Over the course of five weeks (the time it took to measure the *I_c_* after the cable’s preparation), oxygen from the surrounding ambient air might have diffused into the superconductor, resulting in its partial deterioration. We intend to investigate this effect further in the future. The measured values of *I_c_* and *n* are summarized in Table 3.

The cable model was subjected to AC loss measurements, akin to the other samples, at three district frequencies. The results of these measurements, displayed not only in absolute but also in normalized values, are plotted in Figure 13. The normalized values *Q/Q*_0_ are independent of the *I_c_* of the HTS tapes when they are plotted against the normalized magnetic field *H_m_*/*H*_0_. These values were calculated using the measured critical current *I_c_* of the HTS tape and tape width *w_t_* according to the following equation:$$H_{m\;}=\frac{B}{\mu_{0}}\;,\;H_{c}=\frac{I_{c}}{\pi w_{t}}\;,\;Q_{0}=\frac{{\left(\mu_{0}H_{m}\right)}^{2}\pi {w_{t}}^{2}}{\mu_{0}}$$

For comparison, we incorporated three previously acquired AC loss curves in the plots: (1) the unmodified 12 mm wide SP tape, (2) the tape cut to a 4 mm width, and (3) the tape with 15 filaments. We intentionally selected these data to clearly illustrate the impact of cutting and striation on the reduction in AC loss. The sample with 15 filaments was chosen because its filament widths (approximately 750 µm) closely resembled those in the tape wound into a cable (approximately 670 µm). Although the sample with 4 mm width is double the width of the tape used in the cable, it provided the closest matching value for our comparative analysis.

The results in Figure 13a suggest that using narrower tapes is nearly equally beneficial for achieving lower AC losses compared to using wide tapes with a higher number of filaments. Both approaches seem to lead to a similar decrease in AC loss, approximately one order of magnitude. However, in Figure 13b, where normalized values are displayed, the striation approach is significantly more effective in reducing AC loss, while cutting the HTS tape did not genuinely improve the magnetization losses.

Since narrower tapes are more suitable for winding a cable, the model cable in our experiments was wound from the 2 mm wide tape with three filaments. Such tape showed similar results in AC loss reduction as the straight 12 mm wide HTS tape with 15 filaments when their normalized values were compared. At lower magnetic fields, the cable exhibited a lower AC loss, whereas at a higher magnetic field, the situation is reversed. In comparison to the initial state of the HTS tape, the AC loss in the cable was improved six times over. A width of 2 mm is at the border of the practical limits in HTS tape narrowing. Creating striations in such tapes can similarly lead to significant improvements in AC losses as in wide HTS tapes.

The HTS tape employed for the cable was coated using a Ti/AlN/Ti/Ti-Cu/Cu protection system in order to shield the exposed superconductor within the formed grooves. This deposition resulted in a coverage up to 700 nm thin conductive material in the HTS tape’s surface. This thickness is relatively slender and likely exerts minimal influence on the generation of coupling losses. In our upcoming research, we intend to focus more on this protective system.

## 4. Conclusions

This experimental investigation focused on evaluating the *I_c_* retention and magnetization losses in modified HTS tapes. Their modification aimed to reduce the thickness-to-width aspect ratio of the superconductor and was achieved using two approaches: cutting the HTS tapes into narrower widths and creating striations within the superconducting layer. The cutting was performed using a CBN blade and laser ablation. The latter method was used also for the creation of filaments. For both modification approaches, the process parameters were optimized. Specifically, for laser cutting, they were set at 48 W, 80 kHz, and 2000 mm/s, with cooling using compressed air. In the case of striation, the optimized parameters included 6–8 W, 20 kHz, and 500 mm/s, also with cooling using compressed air. The number of repetition cycles was contingent upon the thickness and roughness of the HTS tapes.

Cutting using laser ablation exhibited better *I_c_* retention (98%) in comparison with the mechanical method (82%). When the HTS tapes were narrowed to one-third of their original 12 mm width, they did not exhibit a significant reduction in AC losses, considering the normalized values. Striations created within the superconducting layer are much more effective in reducing AC losses. In comparison to unstriated tape, we measured lower AC losses already one order of magnitude in HTS tapes with eight filaments. However, the *I_c_* retention decreased considerably with a higher number of filaments. While the HTS tapes with 8 filaments (320 µm width) exhibited nearly 100% *I_c_* retention, for the tapes with 15 filaments (750 µm width), it was 91%, and in the tapes with 30 filaments (320 µm width), the *I_c_* retention dropped to only 78%.

A compromise between these results was considered in the design of HTS tape for a cable, which was wound from a 2 mm wide tape with three filaments. The *I_c_* retention of this HTS tape remained stable after undergoing the cutting and striation processes. The cable exhibited a sixfold reduction in AC losses compared to the initial HTS tape.

Overall, the findings of this study provide valuable insights into the modification of HTS tapes to enhance their electrical characteristics, with a particular focus on AC loss reduction, offering a promising path toward the development of pulse magnets made of HTS tapes.

## Figures and Tables

**Figure 1 materials-16-07333-f001:**
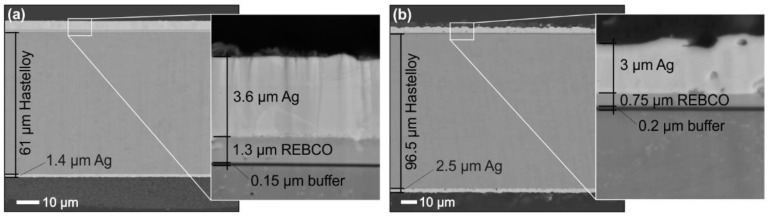
Cross-sections of HTS tapes (**a**) S-Innovations and (**b**) SuperPower with measured layer thicknesses.

**Figure 2 materials-16-07333-f002:**
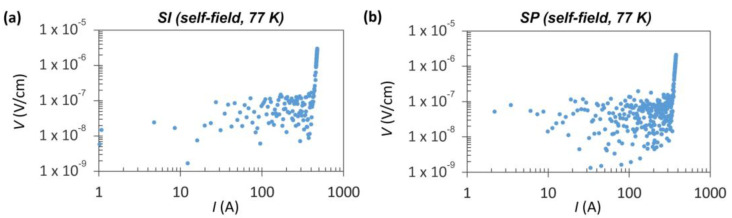
Typical *V-I* characteristics of the investigated HTS tapes: (**a**) S-Innovations, (**b**) SuperPower.

**Figure 3 materials-16-07333-f003:**
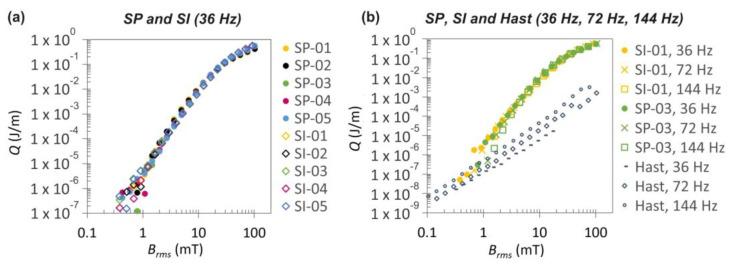
AC losses measured for both S-Innovations and SuperPower HTS tapes in their initial state: (**a**) measurements on five samples for SP and SI at a single frequency, (**b**) measurements on a single sample each for SP and SI at three various frequencies. The graph in part (**b**) is supplemented with AC losses measured in the Hastelloy substrate (Hast).

**Figure 4 materials-16-07333-f004:**
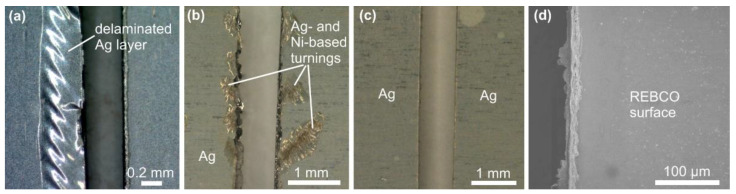
Different quality of cut edges observed in S-Innovations tape when cut using a CBN blade (**a**), and in SuperPower tape when utilizing a diamond (**b**) and a CBN blade (**c**), captured using a light microscope. (**d**) SEM/SE image of a typical crack-free REBCO surface of HTS tape taken from the place shown in (**c**), visible after the Ag layer etching.

**Figure 5 materials-16-07333-f005:**
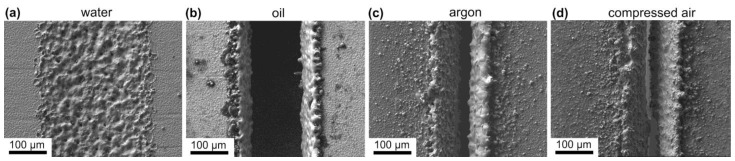
SEM/SE images of cuts performed on SuperPower tapes using laser ablation with the help of different coolants for the samples: (**a**) SP-W1, (**b**) SP-O2, (**c**) SP-A2, (**d**) SP-CA2. Used process parameters are outlined in Table 1.

**Figure 6 materials-16-07333-f006:**
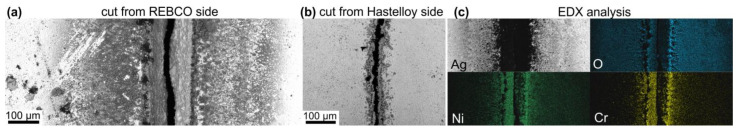
SEM/BSE images of the superconducting side observed in the sample SP-CA2 with material redeposition at the cut edges after laser dividing with compressed air cooling assistance: (**a**) cut from REBCO side, (**b**) cut from Hastelloy side, (**c**) EDX surface distribution of elements from the location shown in (**a**).

**Figure 7 materials-16-07333-f007:**
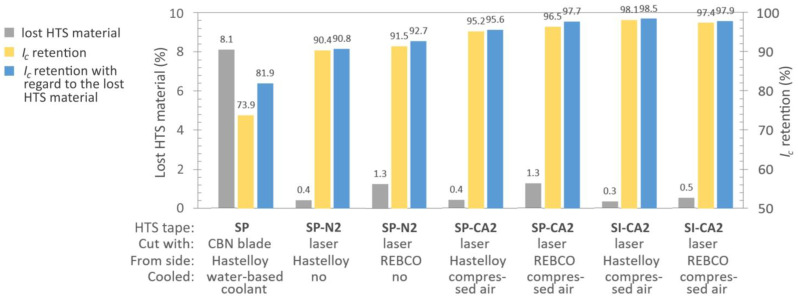
Critical current retention level of both SuperPower and S-Innovations tapes after division of original 12 mm wide tape into three segments, each measuring approximately 4 mm in width. The cutting process involved the utilization of a CBN cutting blade and a laser machine, with cooling and non-cooling options. The positioning of the tapes during cutting varied with respect to the tape’s side. Detail process parameters for laser-cut samples are provided in Table 1.

**Figure 8 materials-16-07333-f008:**
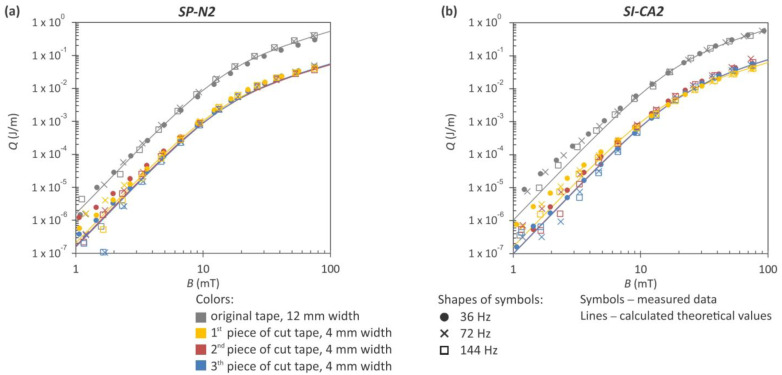
AC magnetization losses as a function of the magnetic field measured for both SuperPower (**a**) and S-Innovations (**b**) original 12 mm wide tapes and after dividing them into three segments of 4 mm width. The experimental data, represented by symbols, were gathered at three distinct frequencies. The solid lines depict theoretical values calculated according to the Brandt–Indebom model.

**Figure 9 materials-16-07333-f009:**
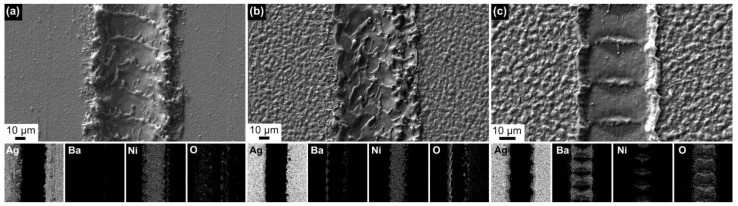
SEM/SE images of grooves prepared in striated HTS tapes using laser ablation, employing process parameters specific to the samples: (**a**) SI-S2, (**b**) SP-S4, (**c**) SP-S3.

**Figure 10 materials-16-07333-f010:**
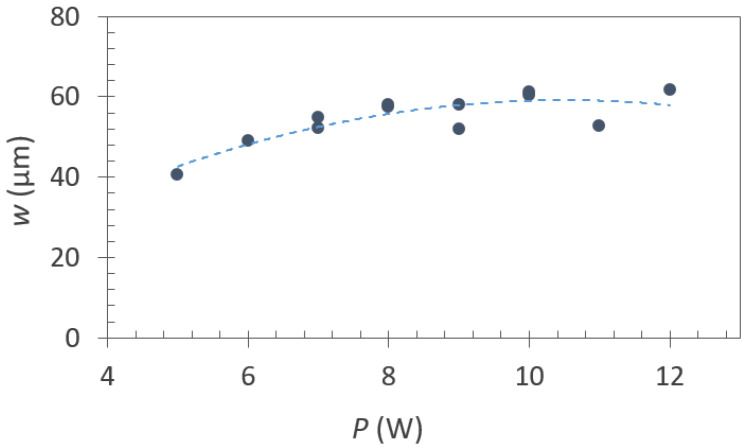
The relationship between groove width and the power of laser device used. All results are from grooves created using two repetition cycles.

**Figure 11 materials-16-07333-f011:**
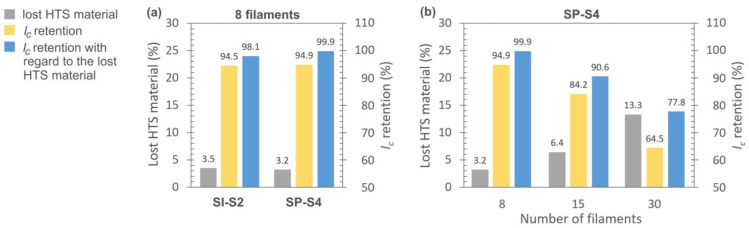
Retention level of critical current in HTS 12 mm wide tapes after laser ablation striation, using compressed air cooling: (**a**) SuperPower and S-Innovations tapes with eight filaments, (**b**) SuperPower tapes with an increasing number of filaments. Details of process parameters for the laser striation of the displayed samples are provided in Table 2.

**Figure 12 materials-16-07333-f012:**
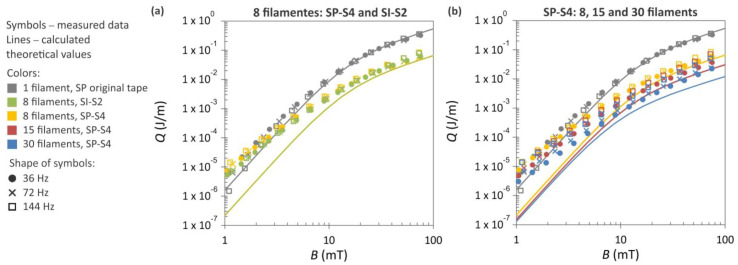
AC magnetization losses depending on magnetic field measured in 12 mm wide HTS tapes: (**a**) SuperPower and S-Innovations tapes with eight filaments, (**b**) SuperPower tape with 8, 15, and 30 filaments. The plots are supplemented with measurements performed on original SuperPower tape. The experimental data, represented by symbols, were gathered at three distinct frequencies. The solid lines depict theoretical values calculated according to the Brandt–Indebom model.

**Figure 13 materials-16-07333-f013:**
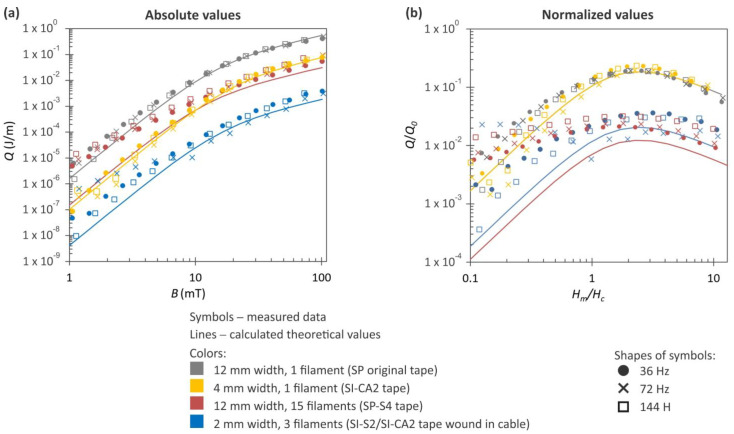
Dependence of AC magnetization losses on magnetic field measured for cut and striated HTS tape wound into a cable. The plots with absolute (**a**) and normalized values (**b**) are supplemented by measurements performed on the original SP tape, the HTS tape cut to a 4 mm width, and the HTS tape with 15 filaments. The experimental data, represented by symbols, were obtained at three distinct frequencies. The solid lines represent theoretical values calculated according to the Brandt–Indebom model.

**Table 1 materials-16-07333-t001:** Process parameters of samples cut using laser ablation.

Sample	Power (W)	Frequency (kHz)	Laser Beam speed (mm/s)	Number of Repetitions	Cooling Medium	Divided
SP-N1	48	80	2000	10	none	no
SP-N2	48	80	2000	20		yes
SP-W1	72	80	1000	100	water	no
SP-W2	100	100	1500	50		no
SP-O1	48	80	2000	20	oil	no
SP-O2	80	80	1000	50		yes
SP-A1	48	80	2000	20	argon	no
SP-A2	52	80	2000	30		yes
SI-A1	48	80	2000	10		no
SI-A2	48	80	2000	13		yes
SI-A3	48	80	2000	15		yes
SP-CA1	48	80	2000	10	compressed air	no
SP-CA2	48	80	2000	20		yes
SI-CA1	48	80	2000	10		no
SI-CA2	48	80	2000	13		yes
SI-CA3	48	80	2000	15		yes

**Table 2 materials-16-07333-t002:** Process parameters of samples striated using laser ablation.

Sample	Power (W)	Frequency (kHz)	Laser Beam Speed (mm/s)	Number of Repetitions	Cooling Medium	Interrupted HTS Layer
SP-S1	1	10	1000	2	None	No
SP-S2	6	20	500	1		Yes
SP-S3	6	20	500	1	Compressed air	No
SP-S4	6	20	500	2		Yes
SP-S5	6	20	500	3		Yes
SI-S1	6	20	500	2		No
SI-S2	10	20	500	2		Yes

**Table 3 materials-16-07333-t003:** Electrical characteristics of HTS tape used for short cable during its modification steps.

	In Initial State of HTS Tape	After Cutting and Striating of HTS Tape	After Winding HTS Tape into Cable
	12 mm width	2 mm width, three filaments	7 mm diameter
*I_c_* (A)	471.0	76.5	67.3
*n*	38.1	35.2	29.5

## Data Availability

Data are contained within the article.

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
