# Peer review of "Optimization of REBCO Tapes through Division and Striation for Use in Superconducting Cables with Low AC Losses"

_materials, 2023, doi:10.3390/ma16237333_

Round 1
Reviewer 1 Report
Comments and Suggestions for Authors
Manuscript number: materials-2705303 Materials MDPI (type of the paper: Article)
TITLE: Optimization of REBCO tapes by its dividing and striating for 2 use in superconducting cables with low AC losses
AUTHORS: Marcela Pekarčíková, Lubomír Frolek , Martin Necpal , Eva Cuninková , Michal Skarba , Simona Hulačová , Filip Ferenčík , Barbora Bočáková
The first review of the manuscript
Overall description of the manuscript
In the manuscript entitled “Optimization of REBCO tapes by its dividing and striating for 2 use in superconducting cables with low AC losses” written by Marcela Pekarčíková, Lubomír Frolek , Martin Necpal , Eva Cuninková , Michal Skarba , Simona Hulačová , Filip Ferenčík , Barbora Bočáková, the author presents extended discussion of the properties of the REBCO superconducting tapes stabilized by Ag. The authors study two commercial available tapes purchased from S-Innovations and SuperPower companies. The materials and methods are well described and it allows to reproduce the results of the measurements by other groups. The authors presents characterization of the tapes in their initial state determining critical currents and AC losses. Next, they investigate effects of cutting of SupperPower tap (narrowing superconductor width).
The paper fits the journal scope quite well. The English language in the manuscript is very good. The paper has 16 pages and includes 39 references (equivalent to 2 pages), 13 figures (about 3 pages), and 3 tables (equivalent to 1 page) – effectively about 10 pages of the main text. The diagrams (figures) are clear, they are essential and their captions are informative. The title clearly and concisely conveys the topic of the article. The abstract quite well describes the content of the manuscript. The findings look correctly. The discussion and conclusions are supported by the results.
In my opinion the manuscript can be published in the present form, however, it would be good to introduce some modification before the final publication. The figures should be larger and more readable. The labels on plots are too small and thus not clearly visible. I believe that suggested improvements can increase the readability and quality of the paper.
To sum up, after these issues will be resolved, I strongly believe the submitted manuscript become suitable for publication in “Materials” MDPI journal as an article provided that the author introduces required changes. The topic of the paper, which is strongly associated with new groups of superconducting materials and its applications, attracts a lot of attention and can be interesting for some groups of scientists and engineers. High temperature superconductivity is still not fully understood phenomenon and any works on this field is desired.
Author Response
We want to convey our sincere gratitude for the valuable comments and suggestions provided by the reviewer.
Comment 1: The figures should be larger and more readable. The labels on plots are too small and thus not clearly visible.
Response:
Thank you for your feedback. We have enlarged several figures to enhance readability. These modifications have been applied to Figure 1, 5, 6, 7, 8, 10, 11, 12 and 13.
Reviewer 2 Report
Comments and Suggestions for Authors
Review on manuscript entitled “Optimization of REBCO tapes by its dividing and striating for use in superconducting cables with low AC losses”.
My team have been working on HTS 2G AC loss estimation, where our approach was based on numerical modeling. Out of doubt, problem considered by authors is relevant. Here a purely experimental investigation is presented. This is very well developed work. Logically written and full of experimental results. I support manuscript publication.
Some small improvements are recommended before publication.
1) Measurement setup can be introduced better in Method section.
2) Figure 7. Left side scale is 50-100%, but largest material lost is 8.1%, probably scale is in 0-10% range? And what is 100% for Ic? Figure 11 looks correct.
3) Some final summarized comparison of all energy losses is missing. Probably a table.
Comments on the Quality of English LanguageGrammar must be polished in my opinion.
Author Response
We want to convey our sincere gratitude for the valuable comments and suggestions provided by the reviewer.
Comment 1: Measurement setup can be introduced better in Method section.
Response:
It seems that you are specifically addressing the calibration-free method, which is not commonly used. Therefore, an addition of a short description of the calibration-free method for AC measurements on page 3 has been noted. For a comprehensive understanding of this method, readers are directed to reference [32] for a detailed explanation.
Comment 2: Figure 7. Left side scale is 50-100%, but largest material lost is 8.1%, probably scale is in 0-10% range? And what is 100% for Ic? Figure 11 looks correct.
Response:
We appreciate your assistance in identifying the error in Figure 7, and the scale has been adjusted accordingly.
100% for Ic was the Ic value measure in initial state of each modified HTS tape (12mm width, not cut, non-striated). This value fell within the range of deviation calculated from the average value of Ic measured on all experimentally used samples, and it was 471±9A for S-Innovations HTS tapes and 367±14A for SuperPower HTS tapes, as mentioned in chapter 3.1.
Comment 3: Some final summarized comparison of all energy losses is missing. Probably a table.
Response:
Thank you for your valuable comment. We have recognized that a summarized comparison is feasible, but only for normalized values, not absolute ones, as the hysteresis loss depends on the conductor's Ic. Our attempt to compare the HTS tapes with significantly different Ic values, led us to include a plot with normalized values in Figure 13. In our opinion, this approach is more suitable than using a table. The introduction of this new graph also resulted in some correction to our conclusions.
Reviewer 3 Report
Comments and Suggestions for Authors
Optimization of REBCO tapes by its dividing and striating for 2 use in superconducting cables with low AC losses by Marcela Pekarčíková et al
Authors have tried to enhance the performance of Ag-stabilized HTS conducting tapes with a focus on reducing magnetization losses. Two approaches were employed viz 1. dividing the tapes into narrower widths and 2. introducing striating at the level of the superconducting layer. The process of laser ablation proved to be an effective method for implementing these modifications. The retention of critical current remained high after cutting, but varied with number of created filaments after striation process. Subsequently, a short cable was wound from cut and striated HTS tape. This cable demonstrated a remarkable two-order-of-magnitude reduction in AC 23 losses compared to the initial HTS tape.
I am happy to recommend this for publication due to the following:
An extensive and application-oriented work that deserves publication.
A clear and lucid development of the paper with adequate results.
Nice data with explanations.
Minor comments:
Authors can introduce a bit of explanation on the details of measurement and its routines as it will help a student reader to understand this.
Can the authors reduce the size of the manuscript as it seems to be bit long and at times boring even for an interested reader. This comment is a general observation and not to be taken otherwise.
Author Response
We want to convey our sincere gratitude for the valuable comments and suggestions provided by the reviewer.
Comment 1: Authors can introduce a bit of explanation on the details of measurement and its routines as it will help a student reader to understand this.
Response:
We acknowledge your comments and have taken note of the need for additional details on measurements, particularly with respect to the not-commonly used calibration-free method. As a response, a brief description of the calibration-free method for AC measurements has been included on page 3. For a more in-depth understanding of this method, readers are directed to reference [32] for a detailed explanation.
Comment 2: Can the authors reduce the size of the manuscript as it seems to be bit long and at times boring even for an interested reader. This comment is a general observation and not to be taken otherwise.
Response:
Certainly, we understand the concern about the length of the manuscript. This feedback is appreciated as a general observation and will be taken into consideration. However, we regret to inform you that reducing the size of the manuscript may not be feasible or possible at this time.
Reviewer 4 Report
Comments and Suggestions for Authors
Manuscript ID: materials-2705303
Title: Optimization of REBCO tapes by its dividing and striating for use in superconducting cables with low AC losses
The present paper is interesting without any doubt. Overall, the manuscript is well-organized and well-written. The linguistic quality of the text is very good, the expressions are clear and concise. The present work contains interesting results that are worth to be published. The obtained experimental results are reliable and seem to be correct. The reported results have been well described and interpreted. Accordingly, the scientific message is conveyed with clarity which makes this publication of great value. Thus, I recommend the present paper for publication. I have a few minor suggestions:
· Usually, current sharing among tapes is critical for stability. This implies that the inter-tape contact resistance (Rc) is a crucial parameter for multi-tape conductors. The contact resistance controls the coupling currents and is able to limit the associated coupling losses in an alternating magnetic field. It would be better to add some graphs on inter-tape contact resistance (Rc) along with a short discussion.
· Is it possible to perform some mechanical testing on the samples?
· I think that some chapters on superconducting cables and tapes included in the following book (https://doi.org/10.1007/978-981-19-1211-5) could be used as references to further enrich the Introduction section with some recent studies.
Author Response
We want to convey our sincere gratitude for the valuable comments and suggestions provided by the reviewer.
Comment 1: Usually, current sharing among tapes is critical for stability. This implies that the inter-tape contact resistance (Rc) is a crucial parameter for multi-tape conductors. The contact resistance controls the coupling currents and is able to limit the associated coupling losses in an alternating magnetic field. It would be better to add some graphs on inter-tape contact resistance (Rc) along with a short discussion.
Response:
We acknowledge the significance of inter-tape contact resistance. However, it is important to note that, in the ongoing experiments detailed in the manuscript, we are still in the initial stages, we mean, we have not yet prepared a cable with more than one tape wound on the electrically isolated core. We intend to address the topic of inter-tape contact resistance in our future experiments.
Comment 2: Is it possible to perform some mechanical testing on the samples?
Response:
Yes, it is possible. Our research group has gained some experience in the mechanical testing of HTS tapes as well as HTS cables. However, in our opinion, this topic is not only important but also quite complex. It could potentially serve as the basis for a new manuscript and we are indeed thinking about it in the future. We would like to prefer to concentrate this manuscript on AC losses of HTS tapes after cutting them in narrower widths and striating.
Comment 3: I think that some chapters on superconducting cables and tapes included in the following book (https://doi.org/10.1007/978-981-19-1211-5) could be used as references to further enrich the Introduction section with some recent studies.
Response:
Thank you for bringing this book to our attention, as it offers a wealth of information on superconductor applications. We have incorporated a reference to it in the introductory chapter.
Reviewer 5 Report
Comments and Suggestions for Authors
This article is of high technological interest. I recommend the authors to perform the following minor revisions:
1) The sentence from lines 129 to 132 is confusing. It could be split into two sentences as the following: "The magnetization losses were measured at 77 K under the effect of an applied external sinusoidal magnetic field. The specific energy, Q, dissipated in each cycle of applied magnetic field and per unit length of the HTS tape samples, was used to quantify the magnetization losses".
2) The paragraph from lines 163 to 170 should come before Table 2.
3) At the end of the first paragraph from section 3.1, the following sentence should be added: "As one may verify from figures 2(a) and 2(b), above the critical current Ic, the applied tension V starts to increase notably due to the increase of the HTS tape samples' electrical resistance".
4) It is better to add the acronym (Hast) after the term "pure Hastelloy substrate", the first time this term appears in the text, in the sentence from line 222 to line 223. So, the new description should be: "Additionally, this graph includes measurements of AC losses for the pure Hastelloy substrate referred as to (Hast).
5) A sentence explaining why in Figure 3(b) the dependence Q(B_rms) is linear for the Hast samples and non-linear for the SP and SI samples, should be added at the end of the second paragraph from section 3.1.
6) The legend from Figure 6, should contain different descriptions for 6(a) and 6(b). The description for 6(a) should refer to the "cut from (RE)BCO side", and the one for 6(b) to the "cut from Hastelloy side".
7) In the conclusions section 5, should be written that when referring to the mechanical blade cutting of HTS tapes: The use of oil as cooling media reduces the material redeposition near the cut edges; The use of compressed air reduces material losses and Ic retention. The use of gas beams implies higher oxidation near the cut edges than when cooled with oil.
8) In the conclusions section 5, should also the written that the cutting by laser ablation reduces notably the material losses with lower Ic retention and the redeposition effect than the mechanical cutting. The authors should also refer to optimum cutting parameters (laser power and pulse duration) when using cutting by laser ablation.
9) A sentence should be added to the conclusions section 5, saying that: "The verified specific magnetization losses Q were about ten times lower (one order lower) for 4mm width filaments resultant from laser ablation cutting than for original 12mm width tapes, this when considering SP-N2 and SI-CA2 samples".
Comments on the Quality of English Language
The quality of English is good requiring minimum edition. Mainly, with respect to the application of commas in the composition of sentences.
Author Response
We want to convey our sincere gratitude for the valuable comments and suggestions provided by the reviewer.
Comment 1: The sentence from lines 129 to 132 is confusing. It could be split into two sentences as the following: "The magnetization losses were measured at 77 K under the effect of an applied external sinusoidal magnetic field. The specific energy, Q, dissipated in each cycle of applied magnetic field and per unit length of the HTS tape samples, was used to quantify the magnetization losses".
Response:
We appreciate your feedback, and we have integrated it into the text.
Comment 2: The paragraph from lines 163 to 170 should come before Table 2.
Response:
The comment has been resolved according to the given suggestions.
Comment 3: At the end of the first paragraph from section 3.1, the following sentence should be added: "As one may verify from figures 2(a) and 2(b), above the critical current Ic, the applied tension V starts to increase notably due to the increase of the HTS tape samples' electrical resistance".
Response:
We have decided not to implement this comment as the information is generally known.
Comment 4: It is better to add the acronym (Hast) after the term "pure Hastelloy substrate", the first time this term appears in the text, in the sentence from line 222 to line 223. So, the new description should be: "Additionally, this graph includes measurements of AC losses for the pure Hastelloy substrate referred as to (Hast).
Response:
Your comment has been addressed as suggested.
Comment 5: A sentence explaining why in Figure 3(b) the dependence Q(B_rms) is linear for the Hast samples and non-linear for the SP and SI samples, should be added at the end of the second paragraph from section 3.1.
Response:
The linear dependence is characteristic of conductive materials, which Hastelloy indeed is. In such materials, AC loss tends to increase with the square power. For superconductors, it rises with the third power, therefore the dependence in non-linear. We have incorporated this explanation into the manuscript on page 6.
Comment 6: The legend from Figure 6, should contain different descriptions for 6(a) and 6(b). The description for 6(a) should refer to the "cut from (RE)BCO side", and the one for 6(b) to the "cut from Hastelloy side".
Response:
The recommendations have been implemented as advised.
Comment 7: In the conclusions section 5, should be written that when referring to the mechanical blade cutting of HTS tapes: The use of oil as cooling media reduces the material redeposition near the cut edges; The use of compressed air reduces material losses and Ic retention. The use of gas beams implies higher oxidation near the cut edges than when cooled with oil.
Response:
This comment is somewhat confusing. Oil was not used as a cooling media for mechanical cutting by blade. We think that the subsequent information provided is not crucial to include it the conclusion.
Comment 8: In the conclusions section 5, should also the written that the cutting by laser ablation reduces notably the material losses with lower Ic retention and the redeposition effect than the mechanical cutting. The authors should also refer to optimum cutting parameters (laser power and pulse duration) when using cutting by laser ablation.
Response:
We acknowledged the importance of incorporating the optimal cutting parameters into the conclusion, and consequently, we have included them including the optimal striating parameters.
Comment 9: A sentence should be added to the conclusions section 5, saying that: "The verified specific magnetization losses Q were about ten times lower (one order lower) for 4mm width filaments resultant from laser ablation cutting than for original 12mm width tapes, this when considering SP-N2 and SI-CA2 samples".
Response:
At the request of another reviewer, we have added a comparison of results using normalized values, illustrated in Figure 13. This approach more accurately reflects the dependency of hysteresis loss on the conductor's Ic. According to these results, the AC losses do not exhibit a decrease. Consequently, we have decided not to include your comment in the conclusion. Instead, we have revised our statement regarding AC losses measured in the cable model.